# Actinide inverse trans influence versus cooperative pushing from below and multi-center bonding

**Laura C. Motta[1,2] & Jochen Autschbach [1] ✉**

Actinide-ligand bonds with high multiplicities remain poorly understood. Decades ago, an effect known as *6p pushing from below* (PFB) was proposed to enhance actinide covalency. A related effect—also poorly understood—is *inverse trans influence* (ITI). The present computational study of actinide-ligand covalent interactions with high bond multiplicities quantifies the energetic contributions from PFB and identifies a hitherto overlooked fourth bonding interaction for 2nd-row ligands in the studied organometallic systems. The latter are best described by a terminal O/N ligand exhibiting quadruple bonding interactions with the actinide. The 4th interaction may be characterized as a multi-center or charge-shift bond involving the *trans* ligand. It is shown in this work that the 4th bonding interaction is a manifestation of ITI, assisted by PFB, and provides a long-sought missing piece in the understanding of actinide chemistry.

Chemical bonding is an essential concept in chemistry. However, the covalent participation of metal 5f atomic orbitals (AOs) in actinide (An) bonding remains comparatively poorly understood. Initial suggestions of An f-shell covalency[1,2] were met with contention[3], because the 5f shell was thought to be radially too contracted to be able to participate in covalent bonding. Although f-shell covalency was correctly asserted more than 70 years ago and has important fundamental as well as practical aspects, its role in actinide chemistry remains challenging to interpret. Fundamentally, the actinides show unusual or unique bonding patterns that continue to push the frontiers of chemistry. As a practical matter, actinide chemistry must be understood and mastered as part of a safe and sustainable nuclear fuel cycle. For example, actinide nitrides have great potential as nuclear-accident tolerant fuel[4].

This article is concerned with a poorly understood—yet extremely important—aspect of An chemistry, viz. unusually high An–ligand bond orders (BOs). It was reported in 2012 that carbon in CUO is triply to quadruply bonded to uranium[5], with one $\sigma$ and two $\pi$ bonds similar to those in uranyl[VI] ([OUO]$^{2+}$), and additionally a weaker so-called rearward $\sigma$ bond. The rearward $\sigma$ bond has not been identified in the isoelectronic uranyl[VI], although it has been suggested that oxo 2s lone pair interactions with uranium may take place[6–8]. It was also speculated that N is quadruply bonded to U in NUN[9].

At the heart of actinide-ligand multiple bonding lie two—also poorly understood—aspects of An chemistry, namely *6p pushing from below* (PFB)[7,8,10–15] and the *inverse trans influence* (ITI)[7,8,16–21]. PFB was initially[10] identified for uranyl[VI], to explain among other aspects the finding that the $\sigma_u$ HOMO, being a bonding linear combination of ligand AOs with 5f AOs, was calculated higher in energy than the $\sigma_g$ involving the metal 6d/7s shells, and higher than the $\pi_{g/u}$ bonding combinations. This was surprising, given that a particularly strong covalent $\sigma$ interaction involving 5f is expected to stabilize the $\sigma_u$ MO[10,22,23], and because the $\sigma_g$ should be higher in energy since it involves the higher-energy 6d metal AOs instead of 5f. The occurrence of a $\sigma_u$ HOMO was then rationalized by an electrostatic repulsion between the ligand lone pairs and the semi-core U 6p shell, potentially accompanied by 6p-5f hybridization. In addition, 6p-oxygen repulsion renders the oxygen a stronger $\sigma$ donor that it would otherwise be.[10] Typically, PFB is assumed for charge-dense ligand anions such as (formal) O$^{2-}$ or N$^{3-}$. It was recently suggested that PFB is also present in certain thorium compounds[24], prompting the question how general this effect truly is.

[1]Department of Chemistry, University at Buffalo, State University of New York, Buffalo, NY 14260-3000, USA. [2]Present address: Department of Marine Chemistry & Geochemistry, Woods Hole Oceanographic Institution, Woods Hole, MA 02543-1050, USA. ✉e-mail: jochena@buffalo.edu

PFB has also been associated with ITI in actinide chemistry. ITI plays a vital role in the successful isolation of terminal organometallic actinide nitrido and oxo multiple bonds. Denning and co-workers[7,8] proposed that a U-ligand bond *trans* to a strong electron-rich bond (e.g., U≡O), is stabilized by parity-allowed 5f-6p mixing, resulting in ITI. The exact role of PFB in the ITI is unknown, however[12,16,18]. The original extended-Hückel theory study of PFB by Tatsumi and Hoffmann[10], and subsequent related theoretical work[11-13,24] did not quantify PFB, and by extension ITI, in terms of energy across different types of compounds. Furthermore, the delocalized nature of the valence canonical MOs has made it difficult to assess the contributions of actinide 6p AOs in larger systems, such that PFB has been evaluated indirectly, for instance via calculations excluding or including 6p in frozen cores[13,16,18,24].

The aim of the present study is therefore two-fold. First, we investigate the extent and energetic contributions from PFB to the electronic structure and its relevance to ITI for a variety of actinide compounds, using modern quantum chemical bonding analyses. The computational strategy harnesses all-electron relativistic Kohn-Sham Density Functional Theory (KS-DFT) and multiconfigurational wavefunction theory (WFT) to identify key orbitals involved in the metal–ligand bonding. The systems are analyzed by the complementary natural bond orbital (NBO) and natural orbitals for chemical valence (NOCV) frameworks, various BO measures, as well as orbital entanglement. Second, PFB is investigated in conjunction with the bonding contributions of the terminal nitrido and oxo 2s lone pairs in the exceptionally covalent $[(R^a)_3N-An-Y]^{n-}$ systems[20,24-27] [$R^a$ = $CH_2CH_2NSi^iPr_3$, Y = N or O, and $n$ = 0, 1, or 2; the compounds have also been labeled as $[An(TREN^{TIPS})]^{n-}$; see Fig. 1a below] and the systems[17] $R-U^{VI}(R^b)_3-O$ [R = Me = $H_3C$ or R = Ph-C ≡ C (Ph = $C_6H_6$), $R^b$ = $N(SiMe_3)_2$; see Fig. 1b below]. The terminal nitrido and oxo ligands were previously assigned as triple or double bonded to the actinide (Fig. 1). We find that these bonds also have considerable rearward $\sigma$ contributions, aided by PFB. It is therefore appropriate to assign quadruple bond character to the terminal nitrido ligands (similar to C≡U in CUO), and even the terminal bonds with O have quadruple bond character, which is exceedingly rare in large organometallic complexes, if not unheard of. The terminal quadruple bond includes a 3-center 4-electron (3c4e) or charge-shift interaction involving the *trans* ligand, and it is facilitated by the covalent, steric, and likely also the electrostatic aspects of PFB, explaining the ITI phenomenon in actinide multiple bonding.

## Results

### Bond orders

NBO-derived BOs from the B3LYP calculations are collected in Table 1. See Supplementary Table 3 for additional BOs according to Mayer[28], Nalewajski-Mrozek (N-M set 3)[29], and Gopinathan-Jug (G-J)[30]. All calculated BOs depend on their underlying partitioning of the density (matrix) and the accompanying definition of the promolecule[31]. Mayer BOs for dative (donation) bonds in systems with metal atoms are often too low to conform to chemical intuition[5,9,25,31,32]. N-M and NBO BOs have been shown to reproduce expected bond multiplicities successfully for a variety of transition metals complexes[31,33], resulting in overall similar bonding patterns. The G-J BOs for our samples are smaller than N-M and NBO because they exclude valence-bond-style ionic contributions, which is likely problematic when considering dative (donation) bonds that are polarized toward the ligands. We focus primarily the NBO-based analysis. The additionally available BO decomposition into contributions from individual Natural Localized Molecular Orbitals (NLMOs)[33], aids the interpretation of the NBO-based BOs.

For the organometallic compounds, the terminal An-N/O BOs are considerably larger than 3, therefore indicating the presence of a fourth bonding interaction. Quadruple bonding between an actinide and a C, N, or O ligand requires the participation of the rearward 2s-rich ligand $\sigma$ lone pair and is therefore almost unheard of. As

mentioned, however, the rearward $\sigma$ bond has been reported for CUO[5]. The present NBO analysis for CUO is in accordance with this previous assignment. We note in passing that the N-M C–UO BO of 4.4 (Supplementary Table 3) should not be interpreted as a quintuple bond, given that the bond is with carbon. Instead, one should view the BO as indicating a particularly covalent quadruple dative bond.

Inspection of individual natural localized molecular orbital (NLMO) contributions to the BOs for the organometallic compounds reveals participation of the terminal N and O rearward $2s_p$ lone-pair NLMO in the bonds. The notation indicates a 2s-rich hybrid with secondary 2p contributions. The $2s_p$ BO contributions are largest for the terminal nitrides (average $0.17 \pm 0.02$), which are close in magnitude to the corresponding value (0.21) for CUO. The covalent contributions of the oxo $2s_p$ (average $0.10 \pm 0.06$) appear weaker because the An-oxo bonds are less covalent overall, but the relative contributions are comparable to those for the terminal N.

The BO analysis also reveals a small but not unimportant participation of the An $6p_\sigma$ AOs, that is, a covalent contribution. The An $6p_\sigma$ BO contributions are greatest for the small molecules (average $0.015 \pm 0.01$, lowest for $ThO_2$) compared to the organometallics (average $0.006 \pm 0.002$). The widespread participation of the An $6p_\sigma$ in the bonding, including the C–U bond in CUO, indicates that PFB is likely to be a general chemical phenomenon, at least among the early actinides. The Th–Cl bond in $(R^a)_3$NTh–Cl has a much smaller BO than most of the other bonds listed in Table 1, as may be expected for a chloride ligand. The absence of a notable rearward $\sigma$ interaction, indicated by the very small $3s_p$ BO contribution, goes along with a negligible Th $6p_\sigma$ contribution for this bond.

### Terminal actinide-ligand triple vs. quadruple bond

Although the BO is calculated to be large for all terminal O/N–An bonds in the studied organometallics, and the involvement of the $2s_p$ ligand lone pair hybrid is evident, further analysis is needed to decide whether these bonds should be assigned as quadruple.

The NBO analysis is designed to determine a best single Lewis structure that can optimally describe the electron density (matrix) upon accounting of some—presumably relatively minor—delocalization. The latter can be described via the contributions from other resonance structures[34,35]. The suitability of a given Lewis structure is quantified by a root-mean-square deviation (RMSD) representing a residual non-Lewis electron number[33-35]. For the closed-shell $(R^a)_3NU^{VI}N$ compound, the optimal Lewis structure as determined by the NBO algorithms (structure 1, in Fig. 1c) features a terminal U≡N and an N-polarized $\sigma(N_{amine}-U)$ bond (83% nitrogen weight). See Fig. 1e, f for NLMO visuals. The analysis (Table 2) shows 1.8 electrons RMSD.

It has been shown previously for a variety of transition metal complexes that a multi-resonance description based on Natural Resonance Theory (NRT) can be much more appropriate than a description based on a single NBO Lewis structure[34-36]. In the present case, NRT analysis revealed, for example, for $(R^a)_3NU^{VI}N$ an important secondary resonance structure featuring a terminal U≡N bond and no $N_{amine}-U$ bond (Fig. 1c, Lewis structure 2). The fourth bond is an N-polarized covalent interaction between the nitride $2s_p$ (89%) and a U $5f6d_{7s6p}$ (an f-d hybrid with contributions from 7s and 6p, Fig. 1). For $(R^a)_3NU^{VI}N$, resonance structures 1 and 2 have the same RMSD individually (1.8 electrons), and according to NRT, the system is best described by strong 45/55 percent resonance. In other words, there is a 3c4e $N_{amine}-U-N_{nitride}$ interaction. The superiority of the resonance model is confirmed by the small residual RMSD (0.09e) in the NRT, meaning the delocalization in the system is essentially only in the $N_{amine}-U-N_{nitride}$ moiety.

Similarly, the NBO-NRT analysis also shows that the resonance model (Table 2) is the best description for the other organometallics, including the PhCC/MeU^{VI}(R^b)_3O compounds (Fig. 1d-f); this picture

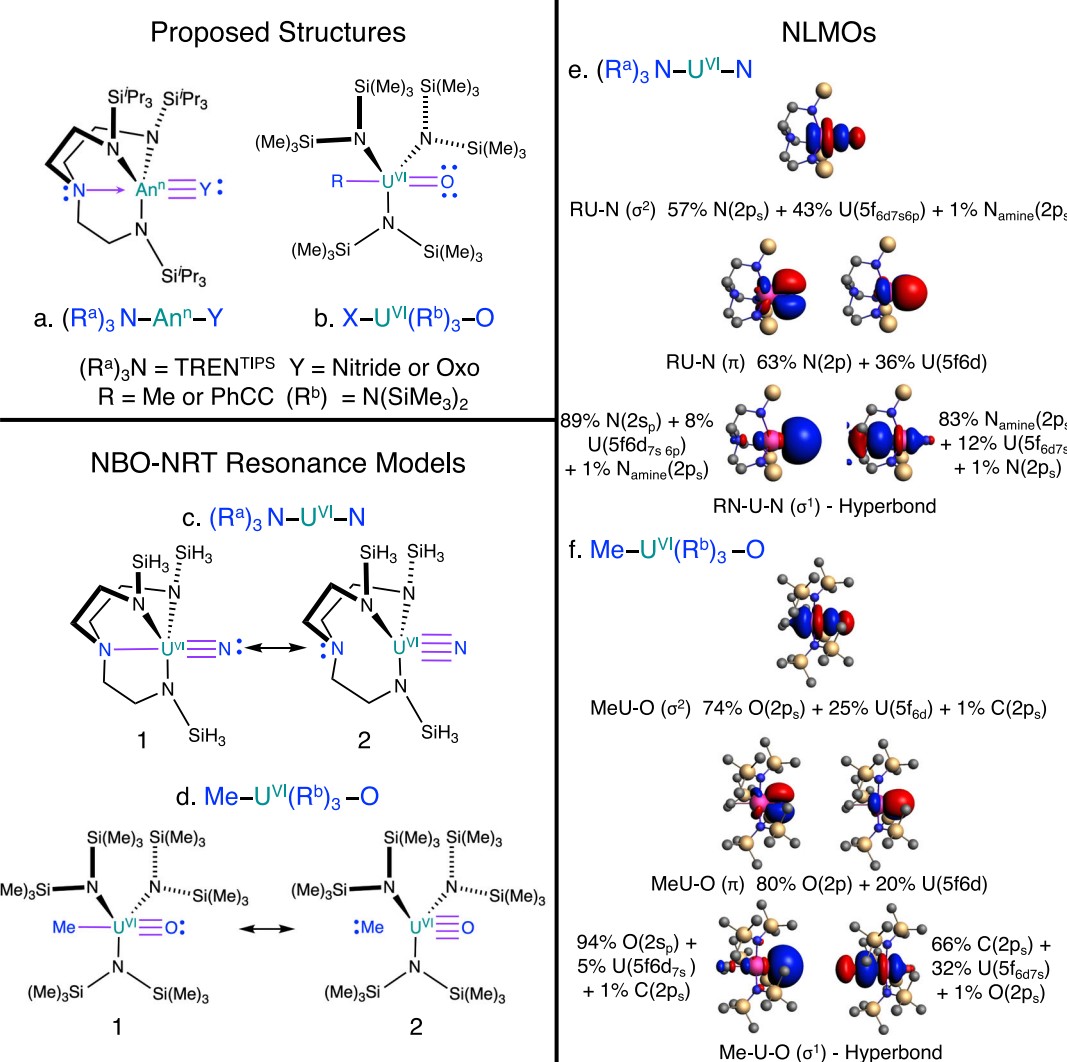

**Fig. 1 | Resonance models for the organoactinide compounds.** Panels **a** and **b** represent proposed structures for the :N(R$^a$)$_3$−An−Y and X−U$^{VI}$(R$^b$)$_3$−O compounds, respectively. Actinides emphasized by green color. The *trans*-interaction is emphasized by purple color, and blue is used for better readability of the panels. The purple arrow represents proposed axial electronic stabilization via ITI by the N$_{amine}$ in the TREN$^{TIPS}$ ligand. Panels **c** and **d** show the dominant Lewis structures 1 and 2 for (R$^a$)$_3$N−U$^{VI}$−N and Me−U$^{VI}$(R$^b$)$_3$−O compounds analyzed by natural bond orbital (NBO) natural resonance theory (NRT). Panels **e** and **f** show isosurfaces (±0.03 atomic units in red/blue) of selected natural localized molecular orbitals (NLMOs, from DFT/B3LYP calculations) for (R$^a$)$_3$N−U$^{VI}$−N and Me−U$^{VI}$(R$^b$)$_3$−O, respectively, with the corresponding AO contributions (some AO hybrids are labeled with subscripts; e.g., 2s$_p$ indicates a 2s-rich hybrid with secondary 2p contributions). Hydrogens are omitted from the structures for clarity.

is supported by the terminal An-N/O and *trans*-An BOs in Table 1. However, for the terminal An-oxo compounds, structure 1 is more dominant given the lower RMSD values relative to structure 2. This is also reflected in the lower BOs for these compounds. In comparison, the uranyl ion is described well by a single Lewis structure ([O ≡ U ≡ O]$^{+2}$), whereas CUO is also resonance stabilized by a 3c4e C−U−O interaction, favoring the C≡O bond. Given that the resonance is of the type X−U ≡ Y: ↔ X: U≡Y (without or with additional X−U bonds), the interaction produces a pronounced U−Y quadruple bond character.

**Stability and classification of the *trans*-An-terminal multi-center interaction**

The BOs and NBO-NRT analyses strongly support the existence of a fourth bonding interaction in the organometallic complexes. However, the analysis does not reveal information about the strength/stability of the 3c4e *trans*−U−terminal-ligand hyperbond. BOs do not strictly correlate with interaction energies, in particular when comparing

different types of chemical bonds[5,37]. To evaluate interaction energies related to the covalency of the 2s$_p$ O and N lone pairs, and the *trans* ligand, NBO-based second-order perturbation energies (Δ$E^{(2)}$) for the donor−acceptor interactions were evaluated and compared to those in [OUO]$^{+2}$, CUO, and NUN. To allow direct comparisons of the Δ$E^{(2)}$ values[33], the X: U≡Y: Lewis structure (structure 1 in Fig. 1) was used consistently. The dominant non-Lewis delocalization interactions are illustrated in Fig. 1c for (R$^a$)$_3$NU$^{VI}$N.

The Δ$E^{(2)}$ stabilization interactions for (R$^a$)$_3$NU$^{VI}$N are illustrated in Fig. 2a and listed in Table 3 for a subset of compounds. For all tested compounds, the energetically dominant interactions (nos. 4 and 5) correspond to donation from the terminal and *trans* ligands lone pairs to formally unoccupied actinide AOs, that is, dative bonding. The strongest interactions are found for the An-terminal bond, in particular for the nitrido systems, in which apparently good overlap and a favorable energy match combine, according to the principles of MO theory, to yield substantial energetic stabilization. Stabilization by the rearward donation to An is also strong for CUO and NUN, in which the

**Table 1 | Calculated (DFT/B3LYP) An–ligand bond orders and An 6p-hole for ionic and organometallic actinide compounds**

| Compound | BL | NBO-NRT BOs | | An-T Covalent NLMO BO decomposition | | | | | $^{An}$6p-hole |
|---|---|---|---|---|---|---|---|---|---|
| | | An–T | Trs–An | $6p_\sigma$ | $2s_{p\sigma}$ | $2p_\pi$ | $2p_{s\sigma}$ | Total | |
| $[U^V\!-\!N]^{2+}$ | 1.75 | 3.0 | – | 0.010 | 0.077 | 1.392 | 0.726 | 2.2 | 0.03 |
| $OTh^{IV}\!-\!O$ $C_{2v}$ | 1.91 | 3.0 | – | 0.003 | 0.019 | 0.550 | 0.422 | 1.0 | 0.04 |
| $OTh^{IV}\!-\!O$ $D_{\infty h}$ | 1.91 | 3.0 | – | 0.006 | 0.025 | 0.504 | 0.442 | 1.0 | 0.03 |
| $[OU^{VI}\!-\!O]^{2+}$ $C_{2v}$ | 1.70 | 2.9 | – | 0.009 | 0.035 | 1.042 | 0.660 | 1.7 | 0.06 |
| $[OU^{VI}\!-\!O]^{2+}$ $D_{\infty h}$ | 1.70 | 2.9 | – | 0.022 | 0.018 | 0.962 | 0.753 | 1.8 | 0.07 |
| $[ONp^{VII}\!-\!O]^{3+}$ | 1.67 | 2.9 | – | 0.023 | 0.020 | 1.262 | 0.824 | 2.1 | 0.09 |
| $[OU^V\!-\!O]^{1+}$ | 1.75 | 2.9 | – | 0.015 | 0.024 | 0.790 | 0.688 | 1.5 | 0.06 |
| $[ONp^{VI}\!-\!O]^{2+}$ | 1.68 | 2.9 | – | 0.020 | 0.022 | 1.022 | 0.790 | 1.9 | 0.08 |
| $NU^{VI}\!-\!N$ | 1.73 | 3.5 | – | 0.016 | 0.110 | 1.318 | 0.948 | 2.4 | 0.06 |
| $OU^{VI}\!-\!C$ | 1.74 | 3.7 | 3.3 | 0.016 | 0.205 | 1.956 | 0.506 | 2.7 | 0.05 |
| $(R^a)_3NTh^{IV}\!-\!Cl$ | 2.70 | 1.1 | 1.0 | 0.000 | 0.017 | 0.234 | 0.213 | 0.5 | 0.01 |
| $[(R^a)_3NTh^{IV}\!-\!N]^{2-}$ | 1.93 | 3.6 | 0.4 | 0.006 | 0.172 | 0.882 | 0.573 | 1.6 | 0.02 |
| $(R^a)_3NU^{VI}\!-\!N$ | 1.80 | 3.5 | 0.5 | 0.009 | 0.195 | 1.458 | 0.873 | 2.5 | 0.03 |
| $[(R^a)_3NU^V\!-\!N]^{1-}$ | 1.83 | 3.5 | 0.5 | 0.006 | 0.171 | 1.276 | 0.777 | 2.2 | 0.02 |
| $[(R^a)_3NU^{IV}\!-\!N]^{2-}$ | 1.83 | 3.6 | 0.3 | 0.006 | 0.157 | 1.098 | 0.695 | 1.9 | 0.02 |
| $(R^a)_3NU^V\!-\!O$ | 1.85 | 3.4 | 0.6 | 0.004 | 0.096 | 0.744 | 0.458 | 1.3 | 0.01 |
| $(R^a)_3NNp^V\!-\!O$ | 1.80 | 3.4 | 0.6 | 0.005 | 0.096 | 0.792 | 0.549 | 1.4 | 0.02 |
| $Me(R^b)_3U^{VI}\!-\!O$ | 1.79 | 3.3 | 0.7 | 0.007 | 0.109 | 0.768 | 0.490 | 1.4 | 0.03 |
| $PhCCU^{VI}(R^b)_3\!-\!O$ | 1.81 | 3.3 | 0.7 | 0.006 | 0.103 | 0.810 | 0.486 | 1.4 | 0.03 |
| $O\!-\!U^{VI}C^*$ | 1.80 | 3.3 | – | 0.009 | 0.004 | 0.681 | 0.580 | 1.3 | 0.05 |
| $(R^a)_3N\!-\!U^{VI}N^*$ | 2.46 | 0.5 | 3.5 | 0.000 | 0.000 | 0.000 | 0.218 | 0.3 | 0.03 |
| $Me\!-\!U^{VI}(R^b)_3O^*$ | 2.34 | 0.7 | 3.3 | 0.003 | 0.000 | 0.000 | 0.639 | 0.7 | 0.03 |

Bond length (BL) in Å for the actinide—terminal ligand (An-T) bond. An–T and *trans*-ligand–actinide (Trs–An) total NBO-NRT bond order (BO; covalent + ionic contributions). Major individual Natural Localized Molecular Orbital (NLMO) contributions to An–T BO. The ionic NBO-NRT BO contributions correspond to a valence bond-type covalent-ionic resonance mixing concept[33]. $C_{2v}$ angle = 120°. $2p_\pi$ = sum of $2p_{mx}$ and $2p_{my}$. *Detailed analysis of the Trs–An bond instead of An–T.

**Table 2 | NRT weights of the Lewis structures 1 and 2, and the associated non-Lewis RMSD errors in the NRT compared to using only a single resonance structure 1 or 2**

| Compound | Weights[a] | | RMSD[b] | | |
|---|---|---|---|---|---|
| | 1 | 2 | Only 1 | Only 2 | NRT |
| $[UOU]^{2+\,c}$ | 99 | – | 0.35 | – | – |
| $CUO^d$ | 75 | 26 | 0.28 | 0.44 | 0.07 |
| $[(R^a)_3NTh^{IV}N]^{2-}$ | 36 | 64 | 2.04 | 1.98 | 0.10 |
| $(R^a)_3NU^{VI}N$ | 45 | 55 | 1.84 | 1.84 | 0.09 |
| $[(R^a)_3NU^VN]^{1-}$ | 48 | 52 | 0.96 | 0.92 | 0.05 |
| $[(R^a)_3NU^VN]^{2-}$ | 34 | 66 | 1.29 | 1.25 | 0.06 |
| $(R^a)_3NU^VO$ | 60 | 40 | 0.88 | 0.92 | 0.05 |
| $(R^a)_3NNp^VO$ | 60 | 40 | 0.87 | 0.92 | 0.05 |
| $MeU^{VI}(R^b)_3O$ | 66 | 34 | 2.99 | 3.34 | 0.08 |
| $PhCCU^{VI}(R^b)_3O$ | 70 | 30 | 5.12 | 6.24 | 0.12 |
| $(R^a)_3NU^{VI}N^e$ | 47 | 53 | 3.31 | 3.31 | 0.00 |

DFT/B3LYP calculations.
[a]Percent weights of Lewis structures 1 and 2 of Figs. 1c and 1d in the resonance stabilized electronic structure according to NRT.
[b]Non-Lewis RMSD (number of electrons).
[c]The dominant resonance structure for uranyl is $O\equiv U\equiv O^{+2}$.
[d]The resonance for CUO is [(1) $C\equiv U\equiv O:\leftrightarrow$ (2):$C\equiv U\equiv O$].
[e]Data for the full experimental $(R^a)_3NU^{VI}N$ crystal structure to serve as comparison with the truncated structure. The overall RMSD numbers are larger due to minor hyperconjugative interactions along the $^i$Pr ligand compared to the truncated $(R^a)_3NU^{VI}N$ compound.

the symmetry of the actinyl ions, the $\Delta E^{(2)}$ values are the same for interaction nos. 4 and 5.

The *trans*-ligand–An-terminal-ligand rearward interaction involves orbitals that are polarized toward the corresponding nitrogen, oxygen, or carbon ligands. There are therefore good reasons to classify the rearward *trans*–An-terminal interaction as a charge-shift (CS) bond[38]. Indeed, using modern VB calculations, it was argued previously that 3c4e bonds in stable (not transient or transition state) species can be classified as CS bonds[38]. CS bonds can also be identified using QTAIM[38], by a small negative or positive electron density Laplacian ($\nabla^2\rho$), densities ($\rho$) $\geq 0.1$, and a negative overall energy density ($H$), at the bond critical point (BCP). In $(R^a)_3NU^{VI}\!-\!N$ the $N_{amine}$–U BCP has a $\rho = 0.1$, $\nabla^2\rho = 0.1$, and $H = -0.03$, and for the U–N BCP we find $\rho = 0.3$, $\nabla^2\rho = 0.04$, and $H = -0.4$. The data are compatible with a CS bond assignment. It is important to note that the QTAIM analysis reflects all of the interactions present in the quadruple terminal bond, not only $2s_p$ rearward bonding.

To further explore the covalent contributions to the An-Terminal bonding, and the stability of the fourth 3c4e/CS interaction, we carried out ETS-NOCV analyses for the organometallic compounds, and compared the results to $[OUO]^{2+}$, NUN, and CUO (Table 3). The ETS-NOCV analysis clearly identifies four bonding contributions to the absolute values of $\Delta E_{orb}$, the stabilization energy that arises from the covalency of the metal−ligand orbital interactions, for most systems: one $\sigma$, two $\pi$, and the additional (weaker) rearward $\sigma$ interaction. The corresponding NOCVs are shown in Fig. 2b–c for $(R^a)_3NU^{VI}\!-\!N$ and $MeU^{VI}(R^b)_3\!-\!O$. Consistent with the other analyses, there is a $\sigma$ bond, two $\pi$ bonds, and the $\sigma$ 3c4e/CS bond. The fourth NOCV clearly displays the multi-center covalent interactions between the *trans* and terminal ligand via U. For the

U is quadruply bonded to the ligand. The NBO $\Delta E^{(2)}$ data indicate that all of the studied actinide compounds are stabilized by the formation of the *trans* 3e4c bond from the resonance of structures 1 and 2. Given

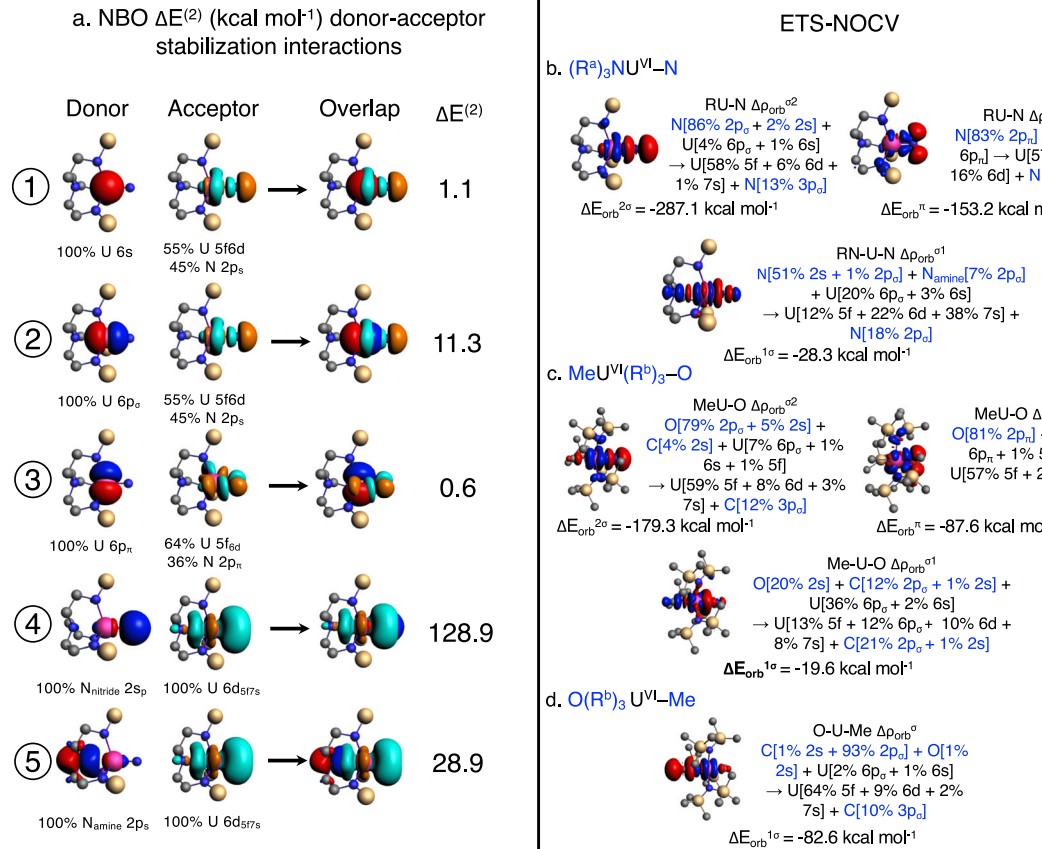

**Fig. 2 | Energetic contributions to metal−ligand bonding.** Left: **a** Natural bond orbital (NBO) representation of dominant non-Lewis delocalization and corresponding $\Delta E^{(2)}$ stabilization (kcal mol$^{-1}$) for $(R^a)_3N-U-N$ shown by the favorable NBO donor−acceptor overlap in each case. Interactions 1 to 3 (circled numbers) correspond to contributions from the semi-core 6s, $6p_\sigma$, and $6p_\pi$ U orbitals; 4 and 5 correspond to donation from the terminal (N) and *trans* $U-N_{amine}$ ligand lone pair to formally empty U and NBOs. NBO isosurfaces at $\pm 0.03$ atomic units in red/blue for donor and cyan/orange for acceptor NBOs. The proposed structure (Fig. 1a) for $(R^a)_3N-U-N$ was used as the Lewis reference for the $\Delta E^{(2)}$ analysis. Right: Energetic contributions $\Delta E_{orb}$ with corresponding natural orbitals for chemical valence (NOCV) analysis (NOCV isosurfaces at $\pm 0.001$ atomic units in red/blue) for the terminal (**b**) $(R^a)_3NU-N$ and (**c**) $Me(R^b)_3U-O$ bonds, and the $\sigma$ *trans* $O(R^b)_3U-Me$ bond (**d**). Hydrogens are omitted from the structures for clarity. DFT/B3LYP calculations.

organometallics, the energetic bond component is the largest for the $\sigma$ bond and the smallest for the 3c4e/CS bond, but the latter is not negligible. Namely, for $(R^a)_3NAn^nN$ compounds, $\Delta E_{orb}$ for the 3c4e/CS bond is larger than that of the known fourth rearward $\sigma$ bond in CUO, and comparable to the covalent energy of the main $[(R^a)_3NTh^{IV}-Cl]$ $\sigma$ bond.

We carried out a subset of ETS-NOCV analyses also for the bonds indicated by the dash in CU−O and $OU^{VI}(R^b)_3$−Me, i.e., to evaluate covalent contributions to the *trans*-An interaction in the 3c4e/CS bond. This analysis is not possible for the $(R^a)_3NAn^nN$ compounds because the contributions of the *trans* $N_{amine}$ cannot be isolated from the TREN$^{TIPS}$ ligand (Table 3). There is a fourth $\Delta E_{orb}$ bonding contribution from the O lone pair in CU−O, although weaker relative to the C lone pair contributions, in agreement with the NBO-NRT resonance analysis. Likewise, the corresponding NOCV for the $OU^{VI}$−Me single $\sigma$ bond illustrates a multi-center interaction composed of C, U, and O contributions.

The combined NBO and ETS-NOCV evidence lead to the conclusion that the *trans*-ligand−An−terminal-ligand interaction in An compounds with high An−ligand bond multiplicities is ITI, facilitated by the terminal and *trans* $2s_p$ AOs. The participation of the actinide 6p semi-core shell is discussed next.

### Analysis of the pushing-from-below mechanism

As discussed earlier, the individual NLMO contributions to the BOs (Table 1) indicate covalent participation of the An $6p_\sigma$ for most of the studied compounds. The $\Delta E^{(2)}$ NBO analysis (Table 3 and Fig. 2; interactions 1−3), also demonstrates clear and energetically important non-Lewis donor−acceptor interactions involving the semi-core $6p_\sigma$ (with the exception of $(R^a)_3NTh^{IV}-Cl$, as noted already). Delocalization of a (partially) filled atomic orbital in a localized orbital framework is covalency, and it creates a partial electron hole in the 6p shell. We find a lesser, but non-negligible extent of 6s and $6p_\pi$ delocalization. The extent of stabilization from $6p_\sigma$ delocalization becomes smaller when going from $(R^a)_3NU^{VI}N$ to $[(R^a)_3NU^{IV}N]^{2-}$.

In a complementary yet consistent picture, the charge-flow channels[39] in $(R^a)_3NU^{VI}-N$ and $MeU^{VI}(R^b)_3-O$ corresponding to four bonding NOCVs (Fig. 1) indicate an outflow of 6p and 6s density to the formally unoccupied 5f and 6d AOs. As expected from the original extended-Hückel theory study of PFB[10], the contributions from the $6p_\sigma$ (~5%) to $\sigma_u$ HOMO are greater than those of the $6p_\pi$ (~1%) to the HOMO-1 and HOMO-2. The largest $6p_\sigma$ (~28%) contributions arise in the $\sigma$ *trans*-U-terminal NOCV, which is not entirely surprising given the comparatively larger radial overlap between the ligand 2s and the $6p_\sigma$ vs. $6p_\pi$ actinide AOs[5,7,8]. Thus, the fourth covalent interaction is a manifestation of PFB facilitating ITI, and it does not require inversion symmetry to be present. As discussed earlier, our results indicate that in addition to 6p-5f hybridization, the covalent participation of the ligand $2s_p$ lone pair in the 3c4e bond is an essential component of ITI.

The covalent aspect of PFB may be accompanied by an energetic destabilization of the ligand $2s_p$ hybrid by electrostatics and by Pauli repulsion with the 6sp shell, facilitating covalent interactions of $2s_p$

**Table 3 | NBO second-order perturbation stabilization energies for the dominant non-Lewis interactions, and the orbital interaction energies $\Delta E_{orb}$ for the dominant bonding interactions shown in Fig. 2a**

| Compound | [a]NBO $\Delta E^{(2)}$ Stabilization interactions | | | | | [b]ETS-NOCV $\Delta E_{orb}$ | | |
|---|---|---|---|---|---|---|---|---|
|  | [c]1 | [c]2 | [c]3 | [d]4 | [d]5 | $\sigma^2$ | $\pi$ | $\sigma^1$ |
| $[OU^{VI}-O]^{2+}$ $D_{\infty h}$ | 2.2 | 14.2 | 1.8 | 29.8 | 29.8 | −320.9 | −168.6 | −16.6 |
| $[OU^{VI}-O]^{2+}$ $C_{2v}$ | 0.8 | 7.9 | 1.6 | 21.0 | 21.0 | −288.9 | −164.7 | −14.0 |
| $OU^{VI}-C$ | 2.1 | 11.7 | 2.4 | 79.7 | 42.9 | −473.0 | −317.2 | −24.6 |
| $(R^a)_3NTh^{IV}-Cl$ | 0.0 | 0.0 | 0.0 | 11.6 | 28.2 | −26.6 | −10.7 | 0.0 |
| $[(R^a)_3NTh^{IV}-N]^{2-}$ | 1.0 | 7.1 | 1.1 | 82.4 | 13.8 | −186.1 | −101.8 | −34.6 |
| $(R^a)_3NU^{VI}-N$ | 1.1 | 11.3 | 0.6 | 128.9 | 28.9 | −287.1 | −153.2 | −28.3 |
| $[(R^a)_3NU^{V}-N]^{1-}$ | 0.4 | 8.0 | 0.4 | 112.1 | 19.6 | −237.0 | −120.2 | −30.0 |
| $[(R^a)_3NU^{IV}-N]^{2-}$ | 0.8 | 7.3 | 0.7 | 73.5 | 24.5 | −169.4 | −83.4 | −27.1 |
| $(R^a)_3NU^{V}-O$ | 0.0 | 6.0 | 0.0 | 79.8 | 22.6 | −144.7 | −70.2 | −19.6 |
| $(R^a)_3NNp^{V}-O$ | 0.3 | 7.6 | 0.0 | 85.6 | 30.0 | −172.7 | −48.4 | −17.4 |
| $MeU^{VI}(R^b)_3-O$ | 0.8 | 10.3 | 0.6 | 62.9 | 61.3 | −179.3 | −87.6 | −19.6 |
| $PhCCU^{VI}(R^b)_3-O$ | 0.6 | 7.3 | 0.0 | 47.3 | 90.4 | −166.8 | −89.3 | −16.7 |
| $CU^{VI}-O^e$ | 1.4 | 9.9 | 0.0 | 42.9 | 79.7 | −209.1 | −77.9 | −13.7 |
| $O(R^b)_3U^{VI}-Me^e$ | 0.0 | 0.0 | 0.0 | 61.3 | 62.9 | −82.6 | −9.1 | – |
| $(R^a)_3NU^{VI}-N^f$ | 1.2 | 10.3 | 0.6 | 81.0 | 21.2 | −287.0 | −152.6 | −27.9 |

DFT/B3LYP calculations.

[a,b]Energies in kcal mol⁻¹.

[c,d] Interactions illustrated in Fig. 2.

[a] NBO second-order perturbation theory donor–acceptor $\Delta E^{(2)}$ stabilization energy.

[b] ETS-NOCV contributions to $\Delta E_{orb}$. The $\Delta E_{orb}$ and $\Delta E^{(2)}$ for the two $\pi$-bonding interactions are equivalent and only one is listed.

[c]$\Delta E^{(2)}$ donor–acceptor interactions involving 6s and 6p An semi-core shells.

[d]$\Delta E^{(2)}$ donor–interaction between the terminal and *trans* ligand lone pair and formally unoccupied An-centered NBOs. For the symmetric small compounds interactions nos. 4 and 5 are identical.

[e]Data for the *trans*–An bond for a selected compounds.

[f]Data for the full experimental $(R^a)_3NU^{VI}N$ structure for comparison.

**Table 4 | Selected axial pairwise steric exchange energies[a] $\Delta E_X$ from NBO analysis for interactions between U 6p/6s and ligand valence NLMOs for the dominant interactions shown in Fig. 2a**

| Compound | NLMO Steric Interactions | $\Delta E_X$ (1) | (2) | (3) |
|---|---|---|---|---|
| $[O-U^{VI}-O]^{2+}$ | U $6p_\sigma \leftrightarrow$ O $2p_s$ | 37.6 | – | – |
| (1) :O ≡ U ≡ O:[b] | U 6s $\leftrightarrow$ O $2p_s$ | 17.4 | – | – |
| (2) O≡U ≡ O: | U $6p_\sigma \leftrightarrow$ O $2s_p$ | 12.3 | – | – |
| (3) :O ≡ U≡O | U 6s $\leftrightarrow$ O $2s_p$ | 0.9 | – | – |
|  | U $6p_\pi \leftrightarrow$ O $2\pi^d$ | 3.5 | – | – |
| $O-U^{VI}-C$ | U $6p_\sigma \leftrightarrow$ C $2p_s$ | 9.6 | 8.0 | 9.1 |
| (1) :C ≡ U ≡ O:[c] | U 6s $\leftrightarrow$ C $2p_s$ | 5.6 | 4.8 | 5.6 |
| (2) C≡U ≡ O: | U $6p_\sigma \leftrightarrow$ C $2s_p$ | 10.2 | 12.8 | 12.7 |
| (3) :C ≡U≡O | U 6s $\leftrightarrow$ C $2s_p$ | 3.3 | 4.0 | 2.9 |
|  | U $6p_\pi \leftrightarrow$ C $2\pi^d$ | 0.3 | 0.3 | 0.3 |
|  | U $6p_\sigma \leftrightarrow$ O $2p_s$ | 36.2 | 36.4 | 40.0 |
|  | U 6s $\leftrightarrow$ O $2p_s$ | 12.2 | 12.3 | 12.9 |
|  | U $6p_\sigma \leftrightarrow$ O $2s_p$ | 9.7 | 9.3 | 1.5 |
|  | U 6s $\leftrightarrow$ O $2s_p$ | 0.2 | 0.1 | 0.5 |
|  | U $6p_\pi \leftrightarrow$ C $2\pi^d$ | 3.6 | 3.6 | 3.6 |
| $(R^a)_3N_{amine}-U^{VI}-N_{nitride}$ | U $6p_\sigma \leftrightarrow N_{nitride}$ $2p_s$ | 34.0 | 32.3 | 23.5 |
| (1) $(R^a)_3N$: U ≡ N:[c] | U 6s $\leftrightarrow N_{nitride}$ $2p_s$ | 11.5 | 12.8 | 8.5 |
| (2) $(R^a)_3N-U≡N$: | U $6p_\sigma \leftrightarrow N_{nitride}$ $2s_p$ | 9.1 | 10.0 | 18.5 |
| (3) $(R^a)_3N$: U≡N | U 6s $\leftrightarrow N_{nitride}$ $2p_s$ | 1.3 | 0.4 | 4.3 |
|  | U $6p_\pi \leftrightarrow N_{nitride}$ $2\pi^d$ | 2.3 | 2.3 | 2.3 |
|  | U $6p_\sigma \leftrightarrow N_{amine}$ $2s_p$ | 8.4 | 8.6 | 8.8 |
|  | U 6s $\leftrightarrow N_{amine}$ $2s_p$ | 4.2 | 3.3 | 4.1 |

DFT/B3LYP calculations. Linear structures for the triatomics.

[a]Energies in kcal mol⁻¹.

[b]The dominant structure for uranyl is O ≡ U ≡ O⁺².

[c]This is not a major resonance structure for CUO or $(R^a)_3N-U^{VI}-N$; it was included for direct comparison with O≡U≡O⁺².

[d]Data for one of the two equivalent U $6p_\pi \leftrightarrow$ ligand$_\pi$ pairwise steric energies is shown.

with the valence 5f and 6d An orbitals. The numerical analysis indeed shows that Pauli repulsion is also an important component of the PFB mechanism. The NBO pairwise steric exchange energies ($\Delta E_X$) associated with the Pauli repulsion between U 6p/6s and the terminal and *trans* ligand valence NLMOs are listed in Table 4 for $[OUO]^{2+}$, CUO, and $(R^a)_3NU^{VI}-N$. Although the Pauli repulsion is more pronounced between the U $6p_\sigma$ and ligand $2s_p$ and $2p_s$ interactions, the participation of the U 6s and $6p_\pi$ shells is not altogether negligible. Overall, it is clear that electrostatic repulsion, Pauli repulsion, and 6p covalency go together in the PFB mechanism.

### Orbital entanglement investigation

The KS-DFT NBO-NRT and ETS-NOCV analyses evidence a terminal An≡L bond via a 3c4e/CS *trans*–An–terminal ligand interaction facilitated by the terminal O or N $2s_p$ AO. Absence of most of the dynamic correlation at the CASSCF level means that the use of NBO-NRT and ETS-NOCV bonding analyses based on this level of multiconfigurational WFT is presently not particularly helpful. To corroborate the rearward lone pair bonding participation by WFT, we therefore conducted a multiconfigurational orbital entanglement analysis[40–46].

Orbital entanglement measures facilitate the qualitative interpretation of the electronic structure in terms of quantum correlation of MOs and have been successful in elucidating bond formation processes[41] and molecular complexation[40]. In particular, the mutual orbital information[40,47,48] indicates entanglement of an orbital pair, hence represent a measure to assess on a qualitative level, independent from the DFT calculations, whether the ligand $2s_p$ and actinide

orbitals are meant to interact. We compare here the orbital entanglement diagrams for OUO⁺², CUO, and $(R^a)_3NU^{VI}N$, shown in Fig. 3.

In the diagrams, the thickness of the connecting lines indicates the extent of entanglement. The diagrams reveal sizable mutual information between the MOs corresponding to the $\sigma$ and $\pi$ U≡L bonds. It is also evident that the ligand lone pair is the least important (least entangled) for $[OUO]^{+2}$. Although for CUO, both the C and O $2s_p$ lone pairs are entangled with formally vacant 5f/6d U orbitals, indicative of the rearward bonding, the C $2s_p$ orbital is more strongly entangled with the LUMO, in qualitative agreement with the preferred C≡U≡O resonance structure. The $(R^a)_3NU^{VI}N$ diagram is qualitatively similar to CUO in this respect, and there is additionally substantial entanglement involving the MOs dominated by $2p_s$ and $2s_p$ AO contributions. Specifically, the $2s_p$ lone pair shares mutual information with a low lying virtual orbital. Closer inspection of the $2s_p$ orbitals shows pronounced mixing with the U 5f, 6d, 6p, and 6s and *trans* $N_{amine}$ $2p_s$ AOs. It appears that the entanglement gives away the multi-center bonding and PFB interactions identified in the DFT calculations.

For the orbital entanglement measurements, we found it necessary to use relatively large active spaces [e.g., MPS(24*e*, 24*o*)]. Previous MPS calculations of neptunium organometallic compounds demonstrated that small CASSCF spaces lead to overlocalization of the 5f shell and an underestimation of the ligand donation (dative bonding)[49,50]. Although (12*e*, 12*o*)[51] and (8*e*, 8*o*)[5] active spaces have been used

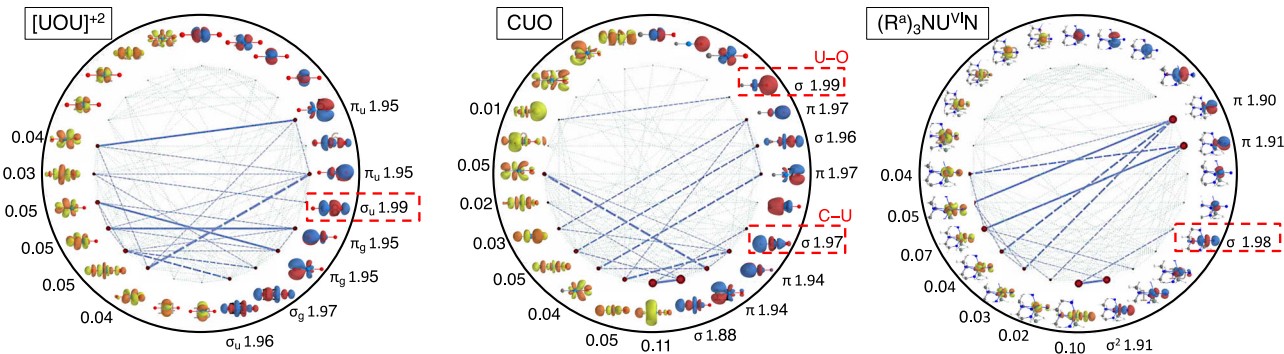

**Fig. 3 | Ground state orbital entanglement diagrams (from MPS(24e,24o) calculations with 12 occupied and 12 virtual orbitals) for the [OUO]$^{+2}$ ion, CUO, and (R$^a$)$_3$NU$^{VI}$N.** Orbitals involving the rearward 2s$_p$ hybrid are highlighted by a red dashed line. MPS natural occupation numbers are given in black for the entangled orbitals. The area of the red circles is proportional to an orbital's single-orbital entropy, while the thickness of connecting lines is proportional to the mutual orbital pair information. Orbital isosurface values are ±0.04 atomic units and rendered in red/blue vs. orange/yellow for high- vs. low-occupancy orbitals, respectively.

previously to investigate the electronic structure of CUO, these active spaces appear to be insufficient to bring about a full picture of the bonding. To describe the C–U and U–O sets of bonds, 4 C–U and 4 U–O (2p$_s$, 2s$_p$ and two $\pi$ each) occupied orbitals are needed. Thus, it is necessary to include at least 8 doubly occupied orbitals (16 electrons) in the active space for CUO. We therefore speculate that the use of smaller active spaces (see Supplementary Fig. 1) in previous research may have obscured the 3-center bonding and ITI interactions in CUO to some degree.

## Discussion

Taken together, the NLMO BO decomposition, $\Delta E^{(2)}$ delocalization energies, and ETS-NOCV bonding analyses paint a clear picture of 6p covalency as well as overlap-driven Pauli repulsion as part of the PFB mechanism. Furthermore, state-of-the-art KS-DFT and multi-configurational WFT-based analyses uncovered a rearward interaction of the 2s$_p$ hybrid ligand lone pair with the metal, previously implicated in the C–U bond in CUO, among a range of actinide compounds. The rearward 2s$_p$ participation facilitates a fourth covalent interaction with terminal nitrido or oxo ligands in several of the studied systems. Actinide 6p$_\sigma$ (and 6s) contributions can be identified in this fourth, rearward interaction, which means that PFB assists, if not enables, a resonance stabilization resulting in the ITI noted for actinide compounds. It also appears that the present study is the first or perhaps one of few reports so far of oxo ligands featuring quadruple bonding interactions. Although the bonds in the studied actinyl ions do not qualify as quadruple, the oxo rearward 2s$_p$ bonding contributions also lead to additional stabilization. The ligand 2s$_p$ bonding contributions are most prominent for the exceptionally covalent organoactinide compounds that were selected for this study, in which we are also able to identify a 3c4e or CS bond involving the *trans* ligand. Overall, the data reveal that the PFB mechanism is a vital component of the bond multiplicity in actinide compounds. It is likely much more prevalent than previously anticipated. (With the latter statement, we echo a conclusion from ref. 24). The present computational study therefore shows that there are still many secrets related to the mystery of 5f covalency that can be uncovered, pushing the boundaries of our understanding of chemical bonding.

## Methods

Absent experimental structures for gas-phase actinyl ions, CUO, and NUN, the geometries for these systems were optimized in linear symmetry with KS-DFT using the Amsterdam Density Functional (ADF) program version 2022[52] with the B3LYP functional[53] and the zeroth-order regular approximation (ZORA) all-electron scalar relativistic Hamiltonian[54,55]. The N–U distance of 1.75 Å for NU$^{2+}$ was taken from ref. 56. For the An(TREN$^{TIPS}$N/O) and PhCC/MeU$^{VI}$[N(SiMe$_3$)$_2$]$_3$

compounds we used the available experimental crystal structures[17,20,24–27]. Given the comparatively large size of the TREN$^{TIPS}$ ligand, truncated model structures replacing $^i$Pr groups by hydrogen were used. Hydrogen positions were optimized as described above. All KS-DFT calculations employed Slater-type orbital (STO) triple-$\zeta$ polarized (TZ2P) all electron basis sets[57]. For the subsequent electronic structure analyses, the eXact two-Component (X2C) Hamiltonian was used[58–60].

NBO analyses[33] were carried out with version 6 of the code included in the ADF suite. For the calculation of the NBO second-order stabilization energies, reference Lewis structures were specified using the CHOOSE keyword[35] to allow direct comparisons between different molecules. Resonant Lewis structures were generated and evaluated using the natural resonance theory (NRT) module in NBO, with high thresholds (20 kcal mol$^{-1}$) to avoid minor intruding hyperconjugative interactions[35,36]. The bonding covalent interactions and the corresponding energy contributions to the total binding energy were evaluated with the extended transition state (ETS) NOCV approach[39] as implemented in ADF. For the ETS-NOCV analyses, the molecules were divided into two ionic fragments by cleaving the terminal bond (e.g., UO$^{4+}$ and O$^{2-}$ for UO$_2^{2+}$, or [U(TREN$^{TIPS}$)]$^{3+}$ and N$^{3-}$ for (R$^a$)$_3$NU$^{VI}$N), to facilitate comparison among different molecules[61]. Pauli repulsion interactions were evaluated for a subset of molecules using the natural steric analysis in NBO[62]. Selected Quantum Theory of Atoms In Molecules (QTAIM) analyses were performed with the Bader module in ADF[63].

To ascertain that the results reported herein are only weakly dependent on the chosen DFT functional, additional bonding analyses were conducted for UO$_2^{2+}$ and (R$^a$)$_3$NU$^{VI}$N with the functionals PBE, PBE0, and TPSSh. Likewise, because several optimized sets of bond lengths for CUO have been reported in the literature[5,51,64–67], we carried out additional analyses for CUO with an X2C/DFT (PBE0 functional) geometry[66], which has slightly shorter C–U and U–O distances of 1.733 and 1.779 Å, respectively, compared to our ZORA/B3LYP bond lengths (1.746, 1.801 Å). Relevant data are provided in Supplementary Tables 1–5, showing that general conclusions can be drawn based on the B3LYP calculations. Given the multi-center nature of the 4th bonding interaction with the rearward 2s$_p$ AO and the general consensus that high An–ligand bond multiplicities are possible in the types of systems studied herein, inclusion of the spin–orbit interaction in the calculations was not deemed to be essential.

In addition to KS-DFT calculations, scalar X2C multi-configurational wavefunction calculations were performed for OUO$^{2+}$, CUO, and (R$^a$)$_3$NU$^{VI}$N with the open-source version of the Molcas program (OpenMolcas)[68–70] at the complete active space (CAS) self consistent field (SCF)[71] matrix product state (MPS) DMRG level[72]. These calculations were used to generate orbital entanglement diagrams,

among other information, with the QCMaquis extension[73–75] of Molcas. These calculations employed Gaussian-type all electron atomic natural orbital relativistic semi-core correlation (ANO-RCC) basis sets[76] in their valence triple-$\zeta$ contractions, except for hydrogen atoms where the double-$\zeta$ contraction was used instead for computational efficiency. The on-the-fly generated auxiliary-basis RICD functionality was utilized for the electron repulsion integrals[77]. Symmetry was not specifically imposed in the wavefunction calculations. Initial orbitals for the orbital-optimizing DMRG runs were obtained from CASSCF calculations with relatively large active spaces [e.g., (16$e$, 13$o$) for CUO, and (12$e$, 13$o$) for ($R^a$)$_3$NU$^{VI}$N]. Series of DMRG calculations were then conducted with active spaces enlarged by subsequent increments of 2 electrons and 2 orbitals, and the orbitals were carefully monitored to ensure that the intended active space was maintained.

Orbital entanglement diagrams were constructed using the AutoCAS software[44]. For the MPS calculations, $m = 1024$ and 15 sweeps were used. The ground state orbital entanglement diagrams were calculated for (14$e$, 14$o$) up to (24$e$, 24$o$) active spaces. Finally, to ascertain that the orbital entanglement diagrams are only weakly dependent on the reported CUO bond lengths, the (24$e$, 24$o$) and (12$e$, 12$o$) orbital entanglement diagrams based on the B3LYP geometry were compared to a (12$e$, 12$o$) diagram based on the PBE0 geometry. The diagrams are provided in Supplementary Fig. 1.

## Data availability

All data generated and analyzed in this study are included in this article, its supplementary information, and the publicly available source files at https://doi.org/10.5281/zenodo.7853933. The zenodo repository includes all ADF and OpenMolcas output files from which the data presented in this study were extracted. Other data related to this study can be obtained from the corresponding author upon request.

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

## Acknowledgements

This work was supported by the U.S. Department of Energy, Office of Basic Energy Sciences, Heavy Element Chemistry program, under grant DE-SC0001136 to J.A. The authors acknowledge the Center for Computational Research (CCR) at the University at Buffalo for providing computational resources and thank Prof. W. H. E. Schwarz for constructive comments on the manuscript.

## Author contributions

L.C.M. performed all calculations and prepared figures, tables, and supplementary data. J.A. designed the project and secured external funding. Both authors analyzed the data and co-wrote the manuscript.

## Competing interests

The authors declare no competing interests.
