## [Peer Review File · Nature Communications]

Actinide Inverse Trans Influence: Cooperative 'Pushing from Below' and Multi-Center BondingReviewers' Comments:

Reviewer #2:

Remarks to the Author:

The manuscript titled Actinide Inverse Trans Influence: Cooperative 'Pushing from Below' and Charge-Shift Bonding by Motta and Autschbach explores in detail previously reported actinide ligand quadruple bonds, extends this assignment to larger coordination complexes, and makes use of state-of-the-art methods to decompose the important but subtle bonding interactions present. Specifically, they show that 2nd-row elements can engage in a sigma type interaction involving actinide 6p orbitals that increases the bond order assignment. They show how this fourth bonding interaction can be understood in terms of two bonding patterns that have been commonly used to understand actinide complexes: pushing from below and the inverse trans influence. I particularly appreciate the careful way in which they connect the two concepts and use a wide variety of tools to do so.

While I think this paper is publishable, I offer some questions in the spirit of improving the manuscript.

--In the discussion of the multireference calculations, I would like the computational details to be clarified a bit.

1. First, the authors note the basis set is ANO-RCC-VXZP. This label is a keyword in OpenMolcas as a shorthand for a specific contraction. It is better practice to report the specific contractions for the atom types.

2. Did the authors use Cholesky Decomposition or any other approximation in computing integrals. If so, they should report it. If not, they may wish to note it.

3. Was symmetry imposed in computing the wavefunctions of the smaller molecules? If so, they should be more specific about how the orbitals were disturbed in the active space. I assume that the linear keyword was used (e.g, Supersymmetry) for NUN and CUO but it should be noted.

--The authors mention on page 5 that other bond orders are computed. They cite prior work about why the Mayer bond orders were not selected but do not discuss the Nalewajski-Mrozek and Gopinathan-Jug bond orders. I think this is a missed opportunity. There is a lot of literature in the community using these bond orders from ADF to assess the "bond order" in actinide ligand bonds. Given the expertise of the authors, can they provide some context for the community? What should a reader make of a Nalewajski-Mrozek bond order greater than 4? Why is there such a large difference between the Gopinathan-Jug and Nalewajski-Mrozek bond orders for say, CUO? I can see that the ordering of the bonds is consistent regardless of method choice, but I'd like to see a paragraph of this in the main paper. Are these differences expected for the authors? Should they concern a reader?

--I am curious why the authors chose to use B3LYP geometries for NUN and CUO when there are CCSD(T) geometries in the literature. I can see why one would wish to use the DFT minimum for the bonding if they had optimized the larger molecules with DFT. Yet, they choose to use the experimental structures there so those bond analyses are not at the B3LYP minimum distances. Especially for the multireference calculations, do you expect any sensitivity to geometry choice? I don't understand why DFT geometry optimizations were not performed for all species. This choice should be explained in the text.

--In the Figures with NLMOs, I'd recommend explaining the notation for the orbital labels (e.g., $U(5f_{6d7s})$). It is explained in the text on page 6, but a reader might first scan the figures.

--Previous studies have used CASSCF calculations to assess bonding with smaller active spaces, like (12,12) for NUN and CUO. Can the authors add a small comparison to such studies? A reader less

familiar with DMRG may not have the experience to compare the approximations required to get larger active spaces with DMRG. Would the authors expect similar bonding would emerge with the smaller active spaces or do you really need the larger active space to see the "fourth" interaction? I see the utility of having the entanglement diagrams, but could the authors do similar analyses on the CASSCF density that they did with DFT? Do they choose not to because they don't expect it will help the argument or because the smaller active space is insufficient?

Reviewer #3:

Remarks to the Author:

Referee report NCOMMS-23-02878-T

This is a very carefully executed and impressive study of bonding in specific actinide compounds. Questions regarding the existence and nature of chemical bonds is, of course, at the heart of chemistry. The study is very well done, from the fundamental level to the specific details. I enthusiastically support publication; I only have a few minor comments.

Figures

I am a bit concerned with the size (font size) of the various text parts and labels in the figures: Assuming that the figure will be reduced in size in a typeset, finished article, will this still be readable?

Minor comments

- P.2, first paragraph: "accident-nuclear" – I think this should be "nuclear-accident"
- Ref. 10: superscripts, subscripts missing
- Ref. 16: n- should be in the superscript
- Ref. 22: ".zeta." – something wrong here with the periods
- P. 7, end of section 3.2: "The long Th–Cl bond length in (Ra)₃NTh Cl explains the absence of Th 6 p σ contributions to the BO." – Would this not be the other way around, i.e. that the long bond length is explained by the absence of the 6p σ ? (Just asking.)
- Figure 2, left panel, perhaps put the "kcal/mol" in brackets: "(kcal/mol)"
- P. 12, "corresponding four bonding NOCVs" – I think there is a "to" missing.
- P. 14, "With the latter statement, we echo a conclusion from Ref. 23" – period missing.

We have revised manuscript NCOMMS-23-02878-T, *Actinide Inverse Trans Influence: Cooperative ‘Pushing from Below’ and Multi-Center Bonding*, in response to the reviewer’s comments and revised the manuscript and supporting information accordingly. Additionally, we shortened the text in some places, to make room with the additions prompted by the reviewers and an added table for selected steric interactions. Formatting, document structuring, and language/grammar instructions according to <https://www.nature.com/documents/ncomms-formatting-instructions.pdf> were applied in the revised manuscript. We also believe that a small change in the manuscript’s title (Charge-Shift → Multi-Center) would be beneficial for the readers. The word count, according to the widely used `texcount` tool (we used \LaTeX for the manuscript) is 5344 including captions and headers, excluding literature citations.

A comparison between the PDF files of the original and the revised submission is provided in file `changes.pdf`.

There were two reviewers, identified in your decision letter as Reviewers 2 and 3. There was no report from a Reviewer 1, and your letter also stated explicitly that *two* reviews were received. We therefore assume that your mentioning of ‘Reviewer 1’ in your decision letter was in reference to the first review, that is, Reviewer 2.

We are grateful to both reviewers for their constructive comments and their careful reading of the manuscript. Below are our responses to the *reviewer comments (in italics)*, along with a description of changes made in the manuscript. The main findings and conclusions from the study remain un-changed. Superscript citations, where provided in this letter, correspond to the bibliography in the revised manuscript.

Reviewer 2 (first of two reviews)

The manuscript titled Actinide Inverse Trans Influence: Cooperative ‘Pushing from Below’ and Charge-Shift Bonding by Motta and Autschbach explores in detail previously reported actinide ligand quadruple bonds, extends this assignment to larger coordination complexes, and makes use of state-of-the-art methods to decompose the important but subtle bonding interactions present. Specifically, they show that 2nd-row elements can engage in a sigma type interaction involving actinide 6p orbitals that increases the bond order assignment. They show how this fourth bonding interaction can be understood in terms of two bonding patterns that have been commonly used to understand actinide complexes: pushing from below and the inverse trans influence. I particularly appreciate the careful way in which they connect the two concepts and

use a wide variety of tools to do so.

While I think this paper is publishable, I offer some questions in the spirit of improving the manuscript.

We thank the reviewer for the positive and encouraging comments and the constructive feedback.

–In the discussion of the multireference calculations, I would like the computational details to be clarified a bit.

We have clarified the descriptions of the multireference calculations, as explained in the following:

1. First, the authors note the basis set is ANO-RCC-VXZP. This label is a keyword in OpenMolcas as a shorthand for a specific contraction. It is better practice to report the specific contractions for the atom types.

The specific basis set contractions that were used are now specified. Please see the amended methods section on page 4.

2. Did the authors use Cholesky Decomposition or any other approximation in computing integrals. If so, they should report it. If not, they may wish to note it.

We used the on-the-fly generated auxiliary basis set approach, activated by the RICD keyword in the SEWARD module, with default parameters. This is now stated in the methods section of the revised manuscript at the bottom of page 4.

3. Was symmetry imposed in computing the wavefunctions of the smaller molecules? If so, they should be more specific about how the orbitals were disturbed in the active space. I assume that the linear keyword was used (eg Supersymmetry) for NUN and CUO but it should be noted.

Symmetry was not specifically imposed in the CASSCF calculations, but we verified that symmetry was preserved in the systems. We opted to not use symmetry because of the procedure we adopted for converging the MPS-DMRG active spaces, which we found easier to accomplish without using spatial symmetry. For example, initial orbitals for the DMRG calculations were obtained from relatively large CASSCF calculation (e.g., CUO = CAS(16,13) and $(R^a)_3NU^{VI}N = CAS(12,13)$). Series of DMRG calculations were then conducted with active spaces enlarged by subsequent increments of 2 electrons and 2 orbitals, and the orbitals were carefully monitored to ensure that the expected active space was maintained. This information was added to the computational details near the top of page 5. In our previous work with DMRG on neptunium complexes, we found that Supersymmetry (SupSym keyword) was necessary to prevent the MPS calculation from ‘rotating’ important orbitals out of the ac-

tive space. However, in the present study such ‘rotations’ could be avoided by the incremental increase of the active spaces as described above.

–The authors mention on page 5 that other bond orders are computed. They cite prior work about why the Mayer bond orders were not selected but do not discuss the Nalewajski-Mrozek and Gopinathan-Jug bond orders. I think this is a missed opportunity. There is a lot of literature in the community using these bond orders from ADF to assess the “bond order” in actinide ligand bonds. Given the expertise of the authors, can they provide some context for the community?

Initially, we did not discuss the other bond orders to keep the manuscript short and because the Nalewajski-Mrozek (N-M) and Gopinathan-Jug (G-J) bond orders provide a picture consistent with what was laid out in the original manuscript. Also, an advantage of using NBO, is the BO decomposition into the major individual Natural Localized Molecular Orbitals. We have now amended the discussion to reflect the Reviewer’s comments in the *Bond orders* section on page 5

What should a reader make of a Nalewajski-Mrozek bond order greater than 4?

Numerical bond orders may deviate from the formal bond orders used by chemists, as is well known.⁶² Some in the actinide community have taken bond orders of slightly above 4 to imply a quintuple bond, but obviously this would make no sense with the involvement of a 2nd-row atom in the bond. As such, we take a numerical bond order > 4 as an indication for a very strongly covalent quadruple bond. This is now stated on page 6 of the revised manuscript. We note also that the N-M bond order definition #3, used by us, has proven to be the most robust in benchmarks⁶² and more robust than G-J or Mayer. This is also now mentioned in the revised manuscript.

Why is there such a large difference between the Gopinathan-Jug and Nalewajski-Mrozek bond orders for say, CUO? I can see that the ordering of the bonds is consistent regardless of method choice, but I’d like to see a paragraph of this in the main paper.

Both the G-J and N-M bond orders were derived using the theory of Valence Bond Indices. The N-M bond orders are typically larger and they are also considered more accurate because the N-M definition includes the ionic contributions from the VB-style covalent-ionic resonance mixing, whereas G-J does not. (The covalent two center part in the N-M bond order is equivalent to the G-J definition.) We now mention this in the discussion of the different bond order measures on page 5 of the revised manuscript.

Are these differences expected for the authors? Should they concern a reader?

Based on the responses provided above, the differences mentioned by the reviewer are not a concern. It is important to note that all of the bond order measures are not far from 4 for the assigned quadruple bonds. As mentioned already, the bond-order discussion has been extended in the revised manuscript.

–I am curious why the authors chose to use B3LYP geometries for NUN and CUO when there are CCSD(T) geometries in the literature. I can see why one would wish to use the DFT minimum for the bonding if they had optimized the larger molecules with DFT. Yet, they choose to use the experimental structures there so those bond analyses are not at the B3LYP minimum distances. Especially for the multireference calculations, do you expect any sensitivity to geometry choice? I don't understand why DFT geometry optimizations were not performed for all species. This choice should be explained in the text.

We were unable to find articles with CCSD(T) geometries for CUO and NUN. Although there are coupled-cluster studies of CUO (refs. 43,44 in the revised manuscript and [Infante & Visscher, J. Chem. Phys. 121 (2004), 5783.] and NUN [Tecmer et al., J. Chem. Phys. 141 (2014), 014107], in those studies the geometries were also optimized using DFT. In Reference 45, the authors summarized all of the published CUO geometries, which included DFT using PBE0 or PW9, SO-CASPT2, and MRSOCISD, but no CCSD(T).

We noticed that the most recent CUO computational studies used X2C/PBE0^{44,45,47} CUO bond lengths (C–U = 1.733 and U–O = 1.779), which are a bit shorter than the ZORA/B3LYP bond lengths calculated by us (C–U = 1.746 and U–O = 1.801). Given the range of reported bond lengths for CUO, we conducted additional DFT NOCV and NBO DFT(PBE0), and CAS(12,12) calculations also based on the X2C/PBE0 structure. There are minor differences between the results based on the different geometries, but the conclusions about the bonding remain the same. This is explained in the revised manuscript on the methods sections page 3 and 4, and added Tables S4 and S5.

For the larger systems, which do have experimental geometries, no harm is done by using those structures, and it is the safe choice for performing the electronic structure analyses. We would not expect DFT structures to be somehow superior to the experimental ones. The reviewer is looking for consistency, which is understandable, but presumably if we had used DFT-optimized structures we would now be prompted for reasons why we didn't use the available experimental structures. In the methods section it is now stated that when available, experimental structures were used, otherwise we used DFT-optimized geometries. We added details about the different reported CUO geometries in the computational methods, and discuss the DFT and CASSCF results in the appropriate sections. We also added additional data in the Supplementary Information comparing the DFT results, and a DMRG

entanglement diagram comparing the CASSCF calculations with both geometries.

–In the Figures with NLMOs, I'd recommend explaining the notation for the orbital labels (e.g., $U(5f_{6d7s})$). It is explained in the text on page 6, but a reader might first scan the figures.

Thank you for the suggestion. We have added the explanations where appropriate. See Figure 1.

–Previous studies have used CASSCF calculations to assess bonding with smaller active spaces, like (12,12) for NUN and CUO. Can the authors add a small comparison to such studies? A reader less familiar with DMRG may not have the experience to compare the approximations required to get larger active spaces with DMRG. Would the authors expect similar bonding would emerge with the smaller active spaces or do you really need the larger active space to see the "fourth" interaction? I see the utility of having the entanglement diagrams, but could the authors do similar analyses on the CASSCF density that they did with DFT? Do they choose not to because they don't expect it will help the argument or because the smaller active space is insufficient?

The reviewer makes an important point. Presently, is not particularly helpful to conduct NBO or NOCV analyses based on the density matrix of a multireference calculation. In CASSCF and DMRG, most of the dynamic correlation is missing, and there is a lack of fully relaxed densities at the CASPT2 level with the codes that we have available. Dynamic correlation is very important for an accurate description of donation (dative) bonding in metal complexes, which we demonstrated in prior work.^{75,76} Therefore, the orbital entanglement diagram is the best approach complementary to DFT for evaluating bonding patterns using multireference wave function methods. It tells us which orbitals 'want' to interact, even if much of the dynamic correlation is missing. NOCV has only been implemented for DFT.

Although, CAS(12,12) and (8,8) active spaces have been used to investigate the electronic structure of CUO, these active spaces are not sufficient. For example, a CAS(12,12) active space cannot accommodate the orbitals needed to describe the C-U and U-O hyperbond. To accurately describe the full set of bonds, 4 C-U (main sigma, rearward sigma, and two pi) and 4 U-O occupied orbitals are needed. Thus, it is necessary to include at least 8 doubly occupied orbitals (16 electrons) in the active space for CUO. As shown in the MPS(24,24) orbital diagram, all 8 occupied orbitals participate in the entanglement. We now mention this in the discussion of the different bond order measures on page 16 of the revised manuscript.

In the SI, we now additionally show the entanglement diagram for CAS(12,12) based on the PBE0 and B3LYP geometries for comparison in the SI (Figure S1). To generate the CAS(12,12) entanglement diagram, we run a DMRG calculation with the CASSCF(12,12)

orbitals using the CIONLY keyword. Using this approach, the MPS(12,12) and CAS(12,12) calculations are essentially the same. We suspect, that the absence of these key U-O orbitals precluded the identification of the C-U-O hyperbond in the past. This is explained in the revised manuscript orbital diagram section.

Regarding the DMRG calculations, we already stated earlier in this letter how the description of the calculations was extended in the revised manuscript.

Reviewer 3 (second of two reviews)

This is a very carefully executed and impressive study of bonding in specific actinide compounds. Questions regarding the existence and nature of chemical bonds is, of course, at the heart of chemistry. The study is very well done, from the fundamental level to the specific details. I enthusiastically support publication; I only have a few minor comments.

We are grateful to the reviewer for the positive comments. We also thank the Reviewer for the corrections, which we all applied.

Figures I am a bit concerned with the size (font size) of the various text parts and labels in the figures: Assuming that the figure will be reduced in size in a typeset, finished article, will this still be readable?

Minor comments - P.2, first paragraph: "accident-nuclear" think this should be "nuclear-accident"

This typo has been fixed.

- Ref. 10: superscripts, subscripts missing

- Ref. 16: n- should be in the superscript

- Ref. 22: ".zeta." something wrong here with the periods

Thank you for pointing out the typos in these references. The references have been edited accordingly.

- P. 7, end of section 3.2: "The long Th-Cl bond length in (Ra)₃NTh-Cl explains the absence of Th 6p contributions to the BO." Would this not be the other way around, i.e. that the long bond length is explained by the absence of the 6p??? (Just asking.)

The bond is strongly polarized toward the ligand, as is typical for chloride, which likely causes the diminished 6p interactions. However, we cannot say whether this is cause or effect, and therefore the statement was revised to say the low BO and low 6p contribution go along

with each other. Please see the amended discussion on page 8.

- *Figure 2, left panel, perhaps put the 'kcal/mol' in brackets: '(kcal//mol)'*

We applied the correction to Figure 2.

- *P. 12, "corresponding four bonding NOCVs?" I think there is a 'to' missing.*

- *P. 14, "With the latter statement, we echo a conclusion from Ref. 23" period missing.*

These typos have been corrected.

Reviewers' Comments:

Reviewer #2:

Remarks to the Author:

After reading the careful response to my prior questions as well as that of the other reviewer, I am very happy to support the publication of this paper. I appreciate the authors' careful responses and willingness to address my questions about their choices. I hope they realize this was done from a sincere place since I know the authors' recognize that there are many ways to approach a modeling study and often one is balancing the impacts of a variety of competing factors. I think a lot can be learned from the discussion of these choices and that this manuscript will be a great contribution to exactly sort of conversation within our community.

Reviewer #3:

Remarks to the Author:

The authors have addressed my comments (that were minor to begin with). As stated before, I enthusiastically support publication.